# Can Tabular Foundation Models Predict Algorithm Runtime Distributions?

Hagverdi Ibrahimli [1]    Steven Adriaensen [1]

## Abstract

Algorithm runtime prediction is a natural testbed for tabular foundation models: it is structured, data-limited, practically important, and requires calibrated uncertainty estimates. Since runtimes can be highly variable and heavy-tailed, we study not only point prediction but full instance-specific runtime distribution prediction. We evaluate TabPFN, a pretrained tabular foundation model performing amortized in-context Bayesian prediction, on established SAT and AI planning benchmarks. TabPFN outperforms classical tabular baselines and specialized neural models, with especially large gains in low-data regimes. At the same time, we uncover a surprising failure mode: on some benchmarks, larger contexts degrade predictive performance. These results establish TabPFN as a strong baseline for empirical runtime modeling and runtime prediction as a challenging benchmark for uncertainty quantification and context scaling in tabular foundation models.

## 1. Introduction

Predicting the runtime of algorithms is a long-standing problem in empirical algorithmics and meta-algorithmics (Hutter et al., 2014), with applications in algorithm selection (Rice, 1976; Kotthoff, 2014; Kerschke et al., 2019), automated configuration (Birattari et al., 2002; Hutter et al., 2009; Schede et al., 2022), and portfolio construction (Gomes & Selman, 2001; Gagliolo & Schmidhuber, 2006; Lindauer et al., 2015). Unlike theoretical complexity analysis, empirical runtime prediction focuses on concrete problem instances and algorithm configurations. In this setting, runtimes often exhibit substantial intrinsic variability and heavy-tailed behavior, even for repeated executions on the same instance (Gomes et al., 2000; Hutter et al., 2014). This motivates predicting full instance-specific runtime distributions (RTDs) rather than only point estimates.

Empirical runtime modeling is a structured-data problem: each observation typically consists of instance features and an observed runtime. Classical tabular methods, especially random forests and Gaussian processes, have therefore remained strong baselines and are widely used in meta-algorithmic systems (Xu et al., 2012; Hutter et al., 2014; Lindauer et al., 2022). Neural approaches have also been proposed for runtime distribution prediction (Gagliolo & Schmidhuber, 2005; Smith-Miles & van Hemert, 2011; Eggensperger et al., 2018; Tuero & Buro, 2021), but despite the broader success of deep learning, they have not clearly displaced classical tabular methods in this domain, plausibly due to the small-data regimes and structured tabular nature of many runtime prediction tasks (Gorishniy et al., 2021).

Tabular foundation models based on in-context learning offer a new perspective on this problem. Prior-Data Fitted Networks (PFNs) perform amortized Bayesian prediction directly from examples provided at inference time, without task-specific gradient-based training (Müller et al., 2022; Hollmann et al., 2023; 2025). This paradigm has enabled neural models to achieve state-of-the-art performance on small tabular datasets, precisely the regime in which conventional deep learning has historically struggled.

In this work, we use TabPFN to answer whether tabular foundation models can predict algorithm runtime distributions. Since TabPFN defines a full posterior predictive distribution, runtime prediction lets us evaluate it not only as a point predictor, but also as an uncertainty-aware model for heavy-tailed, instance-specific targets.

We evaluate TabPFN on established SAT and AI planning benchmarks against classical tabular baselines and specialized neural distributional models. TabPFN achieves state-of-the-art predictive performance, especially in low-data regimes, but also reveals an important limitation: on several benchmarks, larger contexts degrade performance.

Overall, our results establish TabPFN as a strong baseline for empirical runtime modeling and position algorithm runtime prediction as a useful benchmark for uncertainty quantifica-

[1]Department of Computer Science, University of Freiburg, Freiburg, Baden-Württemberg, Germany. Correspondence to: Hagverdi Ibrahimli <ibrahimh@cs.uni-freiburg.de>, Steven Adriaensen <adriaens@cs.uni-freiburg.de>.

*Proceedings of the 2nd ICML Workshop on Foundation Models for Structured Data*, Seoul, South Korea. 2026. Copyright 2026 by the author(s).

tion and context scaling in tabular foundation models.

## 2. Related Work

To our knowledge, we are the first to evaluate a tabular foundation model on algorithm runtime distribution prediction.

**Algorithm runtime distribution (RTD) prediction** is a long-standing problem in empirical algorithmics and meta-algorithmics. Early approaches to this problem relied on parametric regression models and related statistical techniques (Brewer, 1995; Fink, 1998; Howe et al., 1999; Roberts et al., 2007; Leyton-Brown et al., 2009; Xu et al., 2008). Later work demonstrated substantial improvements from nonparametric machine learning models such as random forests and Gaussian processes (Hutter et al., 2014). These methods became the dominant approach in many practical meta-algorithmic systems (Xu et al., 2012; Lindauer et al., 2022). Neural networks have also been explored for runtime distribution prediction (Gagliolo & Schmidhuber, 2005; Smith-Miles & van Hemert, 2011). Most notably, DistNet (Eggensperger et al., 2018) is a feedforward network that is trained to map instance features to the shape and scale parameters of a Lognormal runtime distribution and is optimized end-to-end by minimizing negative log-likelihood. This model achieved strong results across an extensive benchmark suite, outperforming a tree-based baseline; we adopt the same benchmark setting and compare directly against DistNet. Bayes DistNet (Tuero & Buro, 2021) later extended this approach by using Bayesian neural networks and adding support for censored runtime observations.

**Tabular Foundation Models** have recently emerged as general alternatives to classical tabular machine learning. They are pretrained and receive training examples as test-time context to answer queries on unseen data. This in-context learning paradigm was first popularized by large language models (Brown et al., 2020). Prior-Data Fitted Networks (PFNs) are transformers pretrained for in-context Bayesian prediction under a specified dataset prior (Müller et al., 2022). Compared to general-purpose LLMs, PFNs offer a specialized and lightweight route to in-context learning: they are often orders of magnitude smaller, while allowing explicit control over the inductive bias, input representation, and predictive output format. The most prominent example is TabPFN, which has shown strong performance on small-to-medium-sized tabular datasets (Hollmann et al., 2023; 2025), but PFN-style models have since been applied to a wide variety of other problems (Müller et al., 2023; Adriaensen et al., 2023; Dooley et al., 2023; Carstensen et al., 2024; Scheuer et al., 2025). Closest to our setting, KinPFN (Scheuer et al., 2025) applies PFNs to predict RNA folding times, while Carstensen et al. (2024) studies prediction of the latency of neural network architectures, but focuses on point estimates in a narrower setting. Notably, neither study compares its custom PFN approach to TabPFN. While TabPFN is arguably the first, many other tabular foundation models have also been proposed (Kim et al., 2024; Ma et al., 2025; Qu et al., 2025; Zhang et al., 2026); in this first study, we focus on TabPFN because it is widely used, actively maintained, and offers a clear Bayesian interpretation through the PFN framework.

## 3. TabPFN for RTD Prediction

We now describe how we use TabPFN for algorithm runtime distribution (RTD) prediction.

**The RTD prediction problem:** Following Eggensperger et al. (2018), let $A$ be a randomized algorithm and $\Pi_{\text{train}} = \{\pi_i\}_{i=1}^n$ a set of $n$ training instances. Each instance $\pi_i$ is represented by a feature vector $f(\pi_i) \in \mathbb{R}^m$, where $m$ is the number of features, and has $k$ runtime observations $\mathcal{T}(\pi_i) = \{t_j(\pi_i)\}_{j=1}^k$, obtained by running $A$ with different random seeds. Our goal is to learn a model $M$ that, for an unseen instance $\pi$, approximates its instance-specific RTD from $f(\pi)$, i.e., $p_M(t \mid f(\pi)) \approx p(t \mid \pi)$, where $T$ denotes the runtime random variable and $t$ a realization.

More generally, we assume that we are given a training set $D \subseteq \{(f(\pi_i), t_j(\pi_i)) \mid i \in \{1, \ldots, n\}, \; j \in \{1, \ldots, k\}\}$, reducing the problem to a probabilistic supervised regression problem.

**Our TabPFN approach:** The traditional in-weight learning approach would be to optimize a set of model parameters $\theta$ to fit $D$ such that $p_\theta(t \mid f(\pi)) \approx p(t \mid \pi)$. In contrast, our in-context learning approach will take a pretrained model $q$ with frozen parameters that takes as input $D$ and a query feature vector $f(\pi)$ to predict $p_q(t \mid f(\pi), D) \approx p(t \mid \pi)$.

For $q$, we utilize the out-of-the-box v2.5 release of TabPFN, which is reported to scale up to 50,000 samples and 2,000 features (Grinsztajn et al., 2025). We adopt the same data preprocessing steps as Eggensperger et al. (2018). The only additional preprocessing we apply is a log-transformation on the algorithm runtimes $t_j(\pi_i)$; empirical validation confirming the necessity of this transformation for TabPFN is provided in Section C.5 and Figure 6. We use default hyperparameters without tuning or ensembling, obtaining predictions in a single forward pass.

## 4. Experimental Setup

We aim to evaluate the quality of the RTDs predicted by TabPFN and how it varies with context size ($|D|$). In what follows, we briefly describe our baseline models, benchmark data, and evaluation protocol. Further details are provided in Section A.

**Baselines:** We compare our TabPFN approach against classical Gaussian process (GP) and random forest baselines established by Hutter et al. (2014), as well as Dist-Net (Eggensperger et al., 2018), a state-of-the-art neural model for runtime prediction. We omit the Bayes Dist-Net (Tuero & Buro, 2021) extension due to reproducibility issues. We use the defaults proposed in (Hutter et al., 2014; Eggensperger et al., 2018) with manual adjustments for competitive performance (details in Section A.1; ablations in Section C.6 and Figure 7) and additionally tune the random forest with SMAC3 to estimate remaining headroom (Section C.7 and Figure 8).

**Benchmark Data:** We base our empirical evaluation on the datasets introduced by Eggensperger et al. (2018), which comprise seven distinct scenarios spanning propositional Boolean satisfiability (SAT) and AI planning domains. The characteristics of these benchmark instances and their corresponding instance features are summarized in Table 1. For each benchmark, the randomized algorithm was executed with $k = 100$ different random seeds per problem instance, and each resulting pair $(f(\pi_i), t_j(\pi_i))$ is treated as a distinct training sample. We adopt the same data preprocessing steps as Eggensperger et al. (2018). We log-transform runtime targets for all methods except DistNet, for which min–max scaling performed better; an ablation supporting this choice is provided in Section C.4 and Figure 5.

*Table 1.* Benchmark Characteristics.

| SCENARIO | # INSTANCES | # FEATURES |
|---|---|---|
| *clasp_factoring* | 2000 | 102 |
| *saps-CVVAR* | 10011 | 46 |
| *spear_qcp* | 8072 | 91 |
| *yalsat_qcp* | 11743 | 91 |
| *spear_swgcp* | 11182 | 76 |
| *yalsat_swgcp* | 11182 | 76 |
| *lpg-zeno* | 3999 | 165 |

**Evaluation Protocol:** Following Eggensperger et al. (2018), we use 10-fold cross-validation at the instance level, so evaluation occurs on unseen instances. For each training split, we randomly sample contexts of size 32 to 65 536 and repeat this procedure across five random seeds. Beyond the NLLH-focused evaluation from the DistNet study, which is also predominant in PFN-based work, we report CRPS, Wasserstein distance, and Kolmogorov–Smirnov statistics for a fuller assessment of distributional fit (descriptions in Section B). All results are evaluated and reported in the log-space. Finally, to isolate the contribution of instance features, we include a naive TabPFN baseline in which all $m$ features are set to zero.

## 5. Results

We now present our main results. Since results are highly consistent across the seven benchmarks, we focus our discussion on two representative benchmarks. The full set of benchmark results, along with additional experiments, is provided in Section C.

Figure 1a compares all models on two representative scenarios across context sizes and the four distributional fit metrics. In this comparison, TabPFN outperforms all baseline models across all scenario–metric combinations. Differences are largest in the low-data regime, where TabPFN requires up to multiple orders of magnitude less data to match the best score achieved by the closest competitor. TabPFN's superiority across all metrics indicates that it yields better calibrated probability mass estimates compared to the baselines.

Among the evaluated baselines, the GP is the second most data-efficient method. However, its cubic computational complexity causes timeouts under our one-hour wall-clock limit for context sizes $\geq 8192$. In *clasp_factoring*, GP performance also degrades as context size increases, plausibly because repeated observations with identical features but varying runtimes expose heteroscedastic noise that a standard GP with a single global noise parameter cannot capture. Surprisingly, in low-data regimes, feature-ablated TabPFN-Naive sometimes outperforms traditional baselines, likely because existing methods are unstable in this setting whereas TabPFN can still exploit the prior-induced inductive bias learned during pretraining. Results for the remaining five scenarios are consistent with these findings; see Section C.1 and Figure 2.

Despite these advantages in low-data regimes, TabPFN's predictive performance begins to degrade beyond a context-size threshold (e.g., 2048 samples for *clasp_factoring*). Because our datasets contain relatively few unique instances, larger contexts include increasing numbers of repeated feature vectors paired with different runtimes. We hypothesize that such repetition may be underrepresented in TabPFN's pretraining distribution and could contribute to the observed degradation. To examine this hypothesis, we repeated the experiment while limiting the context to at most one runtime sample per instance. This deduplication intervention largely mitigated the degradation trend across the evaluated metrics, as illustrated for *lpg-zeno* and *clasp_factoring* in Figure 1b. Results for all benchmark scenarios are provided in Section C.2 and Figure 3.

## 6. Limitations and Future Work

Our results establish TabPFN as a strong off-the-shelf baseline for empirical runtime modeling and position algorithm runtime prediction as a challenging benchmark for tabular

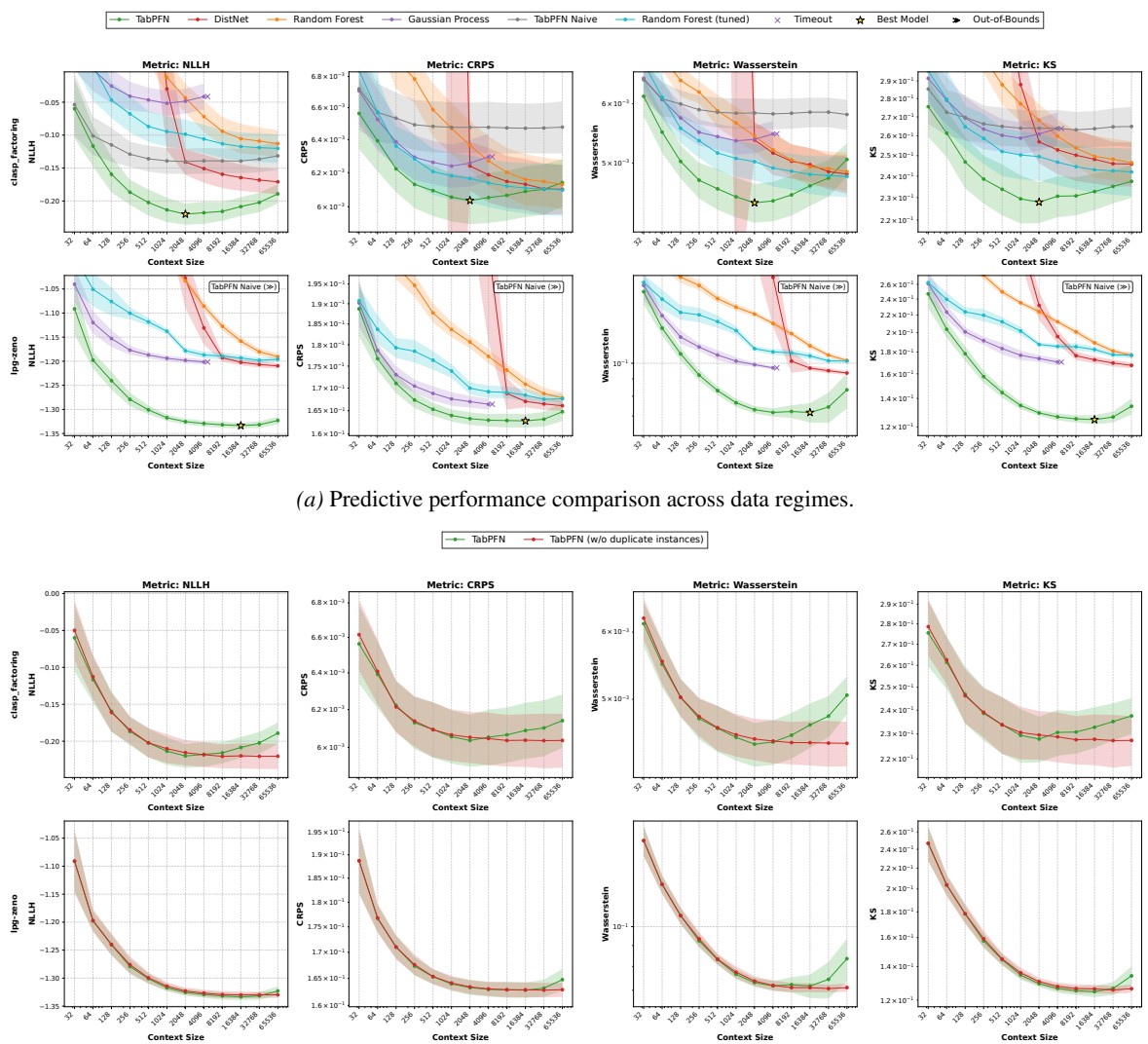

*(a)* Predictive performance comparison across data regimes.

*(b)* Standard vs. deduplicated TabPFN (at most one runtime per instance).

*Figure 1.* **Predictive performance across data regimes.** Both subfigures report performance across context sizes for *clasp_factoring* and *lpg-zeno* (rows) using NLLH, CRPS, Wasserstein distance, and the KS statistic (columns). Solid lines show mean scores over 10-fold instance-level cross-validation with five random seeds per fold; shaded regions indicate $\pm 1$ standard deviation. Lower is better. Stars in subfigure (a) mark the best model–context-size combination for each benchmark and metric.

foundation models more broadly. The simplicity of our setup is also its main limitation: we evaluate TabPFN as-is, without runtime-specific priors, task-specific training, architectural changes, or hyperparameter tuning. Similarly, our classical baselines mainly use established defaults rather than extensive per-task tuning, with a SMAC3-tuned random forest providing an indication of tuning headroom. Although tuning improves the random forest, it closes only a limited portion of the gap to TabPFN, leaving open the gains achievable through systematic tuning on either side. Runtime prediction therefore emerges as both an application and a stress test, exposing open challenges in heavy-tailed uncertainty quantification and context scaling. While context deduplication mitigates the predictive degradation at larger

contexts, the steep computational and memory costs detailed in Section C.3 remain a bottleneck. Future work should address these computational scalability issues through context compression and alternatives to dense attention, should analyze the impact of hyperparameter tuning for both TabPFN and competing baselines, and should evaluate whether other tabular foundation models exhibit similar strengths and failure modes. Looking beyond this benchmark, the most exciting direction is to turn runtime prediction into a building block for downstream meta-algorithmics: PFNs that ingest richer algorithm and configuration data, handle censored observations, and learn across algorithms and domains could use contextual experience to allocate computation, select algorithms, configure solvers, and steer search.

## Acknowledgements

The authors would like to thank Katharina Eggensperger from the Lamarr Institute at TU Dortmund University for her valuable discussions and insights. Hagverdi Ibrahimli acknowledges financial support from the Deutschland-stipendium. Furthermore, the authors acknowledge funding by the state of Baden-Württemberg through bwHPC and the German Research Foundation (DFG), supporting project number 455622343 (bwForCluster NEMO 2).

## Impact Statement

This paper presents work whose goal is to advance empirical runtime modeling and meta-algorithmics. By improving predictions of algorithm runtime distributions, our work may support more informed algorithm selection, portfolio construction, and resource allocation decisions, especially in settings where computational cost is uncertain. As with other automated decision-support tools, such predictions should be used with awareness of their limitations, including possible miscalibration or distribution shift. We do not identify further direct societal consequences that warrant specific discussion here.

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

# A. Detailed Experimental Setup

This appendix details the experimental setup used throughout our empirical evaluation. We first specify the baseline runtime distribution predictors in Section A.1, covering the neural DistNet model as well as the random forest and Gaussian process baselines, and describe their distributional assumptions, model configurations, and procedures for obtaining predictive runtime distributions. We then describe the benchmark data, preprocessing pipeline, and evaluation protocol in Section A.2, including the SAT and AI planning scenarios, the handling of individual instance-seed evaluations, the instance-level cross-validation and subsampling scheme, and the target transformations used by the different baselines.

## A.1. Baselines

We compare our TabPFN approach against DistNet (Eggensperger et al., 2018), a neural approach designed for algorithm runtime distribution prediction, as well as classical Gaussian process and random forest baselines established by Hutter et al. (2014).

**DistNet.** We include DistNet, a fully connected feedforward neural network previously demonstrated to be highly competitive for this specific task. While Eggensperger et al. (2018) explored various parametric forms for modeling the empirical RTDs, the Lognormal performed best, which is the variant we used. The network is therefore designed to jointly predict the two parameters of this distribution: the shape and the scale. The architecture is compact, consisting of two hidden layers with 16 neurons each, relying on the $\tanh$ activation function, and utilizing an output layer with an $\exp(\cdot)$ activation to guarantee strictly positive parameter predictions. We optimize the network by directly minimizing the negative log-likelihood (NLLH) in an end-to-end fashion. Training relies on stochastic gradient descent (SGD) with a batch size of 16 to reduce training data correlation, alongside an initial learning rate of $10^{-3}$ that exponentially decays to $10^{-5}$. Given the risk of large gradients inherent to NLLH optimization, where slightly suboptimal parameter predictions can yield likelihoods close to zero, we employ gradient clipping, batch normalization, and an $L_2$-regularization penalty of $10^{-4}$. Additionally, early stopping is applied to mitigate overfitting on the training set.

**Random Forest.** We adopt the random forest baseline specified by Hutter et al. (2014) for algorithm runtime distribution prediction. By predicting the parameters of an underlying Gaussian distribution $(\mu, \sigma^2)$ over the log-scaled runtime space, this model effectively assumes a Lognormal RTD in the original space. To quantify predictive uncertainty, the model calculates the overall predictive mean as the average of the individual trees' means. Crucially, following the law of total variance, the total predictive variance combines the variance across the means predicted by individual trees (indicating ensemble disagreement) and the average variance within each tree. The ensemble consists of 50 unpruned regression trees grown deeply until each leaf node contains a maximum of 5 data points. At each split, the model considers a random 50% subset of the features and selects a split point uniformly at random from the interval between two continuous values, yielding smoothly varying variance estimates. Each leaf node stores both the empirical mean and variance of its contained data, with the minimum variance strictly bounded to $10^{-6}$ to prevent overly deterministic predictions.

**Gaussian Processes (GP).** We selected a Gaussian process (GP) model as an additional data-efficient baseline. Like the random forest, the GP predicts the parameters of an underlying Gaussian distribution $(\mu, \sigma^2)$ over the log-scaled runtime space, thereby assuming a Lognormal RTD in the original space. To construct an expressive prior that captures the complexities of empirical runtime data, we employ a composite kernel of the form `ConstantKernel × Matern +` `WhiteKernel`. Specifically, we use a Matérn kernel ($\nu = 1.5$) to model the underlying once-differentiable performance landscape, scaled by a constant kernel to learn the signal variance. To account for the aleatoric uncertainty arising from the intrinsic stochasticity of the algorithms across different random seeds, we further incorporate a white noise kernel. All kernel hyperparameters are optimized within bounds of $[10^{-5}, 10^5]$, and the log-marginal likelihood is maximized with three random restarts to prevent the model from converging to local optima.

## A.2. Benchmark Data, Preprocessing, and Evaluation Protocol

We base our empirical evaluation on the datasets introduced by Eggensperger et al. (2018), which comprise seven distinct scenarios spanning propositional satisfiability (SAT) and artificial intelligence (AI) planning domains. The characteristics of these benchmark instances and their corresponding instance features are summarized in Table 1.

To gather the runtime data, each randomized algorithm was executed with $k = 100$ different random seeds per problem instance. We follow the preprocessing protocol of Eggensperger et al. (2018): instances with failed or timed-out runs and constant feature vectors are removed; missing feature values are imputed using the feature-wise median. For all reported

experiments, we perform 10-fold cross-validation at the problem-instance level before flattening the training data into individual instance-seed evaluations, ensuring that test instances remain strictly disjoint from the training instances in each fold. We then flatten the training splits such that each instance-seed pair $(f(\pi_i), t_j(\pi_i))$ becomes a distinct sample, increasing the effective training set size by a factor of $k = 100$. Based on the respective training split, we then normalize all continuous features to a mean of 0 and a standard deviation of 1, and remove any constant feature columns. From the resulting training set, we randomly sample a subset of data points (of a size specified by the context size) for model fitting; to ensure robustness, we repeat this sampling and model fitting across five independent random seeds within each fold to account for stochasticity.

Crucially, we apply a log-scaling transformation to the target runtime values for TabPFN, the random forest, and the GP baselines. Since algorithm runtimes vary substantially, often by orders of magnitude, this logarithmic transformation allows baseline models that output standard Gaussian predictive distributions to effectively model the Lognormal RTDs of the original, unscaled runtime space. In contrast, for DistNet, we retain the min–max-scaled target space originally proposed by Eggensperger et al. (2018), as our ablation study confirmed that this scaling choice outperforms log-scaling on average for this specific neural network architecture (see Section C.4 and Figure 5).

## B. Evaluation Metrics

This appendix describes the evaluation metrics used to quantify the quality of the predicted runtime distributions. Because empirical algorithm runtimes are heavy-tailed and often span multiple orders of magnitude, we conduct all metric evaluations in the log-transformed runtime space and follow the 10-fold instance-level cross-validation protocol described in Section A.2. We first introduce the common notation used throughout the metric definitions. We then provide formal definitions for the four distributional metrics used in our evaluation, namely the scale-adjusted negative log-likelihood in Section B.1, the continuous ranked probability score in Section B.2, the 1-Wasserstein distance in Section B.3, and the Kolmogorov-Smirnov distance in Section B.4.

Given an instance $\pi$ with $k$ runtime observations $\mathcal{T}(\pi) = \{t_1(\pi), \ldots, t_k(\pi)\}$ generated using different random seeds, let $z_j(\pi) = \log(1 + t_j(\pi))$ denote the corresponding log-runtimes. Equivalently, because $T$ denotes the runtime random variable, $Z = \log(1 + T)$ denotes the corresponding log-runtime random variable. Accordingly, we let $\mathcal{Z}(\pi) = \{z_1(\pi), \ldots, z_k(\pi)\}$ denote the set of log-runtime observations for instance $\pi$.

A predictive model $M$ outputs an estimated log-runtime distribution for $\pi$. Let $p_M(z \mid f(\pi))$ denote the predicted probability density function (PDF) and $F_M(z \mid f(\pi))$ denote the predicted cumulative distribution function (CDF) in the log-space. The $k$ empirical observations define a stepwise empirical CDF, given by:

$$\hat{F}_\pi(z) = \frac{1}{k} \sum_{j=1}^{k} \mathbb{I}(z_j(\pi) \leq z) \tag{1}$$

where $\mathbb{I}(\cdot)$ is the indicator function.

### B.1. Negative Log-Likelihood (NLLH)

The standard NLLH evaluates the pointwise predictive density assigned to the true observations. However, as Eggensperger et al. (2018) note in their paper, directly aggregating raw likelihoods across diverse instances is problematic: "easy" instances naturally exhibit narrow distributions with high density peaks, whereas "hard" instances possess wide distributions with inherently lower density peaks. To prevent instances with narrow variance from disproportionately dominating the aggregate loss, we adopt the scale-adjusted formulation introduced by Eggensperger et al. (2018).

Adapted to our log-space evaluation, the scale-adjusted NLLH for a single instance $\pi$ is defined as:

$$\text{NLLH}(\pi) = -\left(\frac{1}{k} \sum_{j=1}^{k} \log p_M(z_j(\pi) \mid f(\pi))\right) - \log \max_{j \in \{1, \ldots, k\}} z_j(\pi) \tag{2}$$

The subtracted logarithmic maximum term acts as an instance-specific bias correction. This adjustment standardizes the likelihood bounds across instances with drastically different runtime magnitudes, ensuring that the model is evaluated on the structural fit of the distribution rather than the inherent scale of the problem.

## B.2. Continuous Ranked Probability Score (CRPS)

The CRPS is a strictly proper scoring rule that generalizes the Mean Absolute Error (MAE) to probabilistic forecasts. It simultaneously evaluates both calibration (statistical consistency) and sharpness (concentration of probability mass). In our problem setup, the ground truth is a set of $k$ observations rather than a single scalar. Consequently, we compute the instance-level CRPS as the expectation of the standard single-observation CRPS over the empirical distribution:

$$\text{CRPS}(\pi) = \frac{1}{k} \sum_{j=1}^{k} \int_{0}^{\infty} \left( F_M(z \mid f(\pi)) - \mathbb{I}(z_j(\pi) \leq z) \right)^2 dz \tag{3}$$

This metric heavily penalizes probability mass that is assigned far from the observed log-runtimes. Evaluating this integral in the log-space is mathematically advantageous: it translates the physical distance penalty from absolute raw seconds into a relative error (log-seconds), maintaining stability when evaluated across instances spanning vastly different orders of magnitude.

## B.3. 1-Wasserstein Distance

The 1-Wasserstein distance, frequently referred to as the Earth Mover's Distance, quantifies the global physical effort required to morph the predicted probability distribution into the empirical ground-truth distribution. For one-dimensional distributions, this resolves exactly to the $L_1$ integral of the absolute differences between their respective CDFs:

$$\mathbf{W}_1(\pi) = \int_{0}^{\infty} \left| F_M(z \mid f(\pi)) - \hat{F}_\pi(z) \right| dz \tag{4}$$

In the context of algorithm runtime prediction, the Wasserstein distance provides a highly interpretable physical metric. It represents the average absolute displacement (in log-seconds) necessary to perfectly align the model's predicted probability mass with the $k$ empirical runtime observations. Unlike the NLLH, it gracefully handles disjoint probability supports and provides a stable, global assessment of how well the predicted distribution's location and scale match reality.

## B.4. Kolmogorov-Smirnov (KS) Distance

The KS distance evaluates the worst-case local miscalibration of the predictive distribution. It is defined as the supremum of the absolute vertical difference between the predicted and empirical CDFs:

$$\text{KS}(\pi) = \sup_{z \geq 0} \left| F_M(z \mid f(\pi)) - \hat{F}_\pi(z) \right| \tag{5}$$

Unlike the CRPS and the 1-Wasserstein distance, the KS distance does not integrate over the physical log-runtime space ($dz$). Consequently, it isolates purely structural probability misalignment and is completely invariant to the physical magnitude of the prediction error. This makes the KS distance an excellent supplementary metric for identifying the single threshold where the model's confidence deviates most severely from the true empirical frequency.

# C. Additional Experimental Results

This appendix reports additional experimental results that complement the main empirical evaluation. All experiments follow the 10-fold instance-level cross-validation protocol described in Section A.2, with all distributional metrics evaluated in the log-runtime space as defined in Section B. Whereas Figure 1a reports predictive performance for two representative scenarios in the main text, Section C.1 provides the corresponding full comparison across all seven benchmark scenarios, all data regimes, and all four distributional metrics. We then evaluate the effect of repeated runtime observations attached to the same instance-feature vector on TabPFN in Section C.2, followed by computational profiling of fit time, prediction time, amortized prediction time, and memory scaling in Section C.3. Finally, we report targeted ablations for the implementation choices used in the main experiments, including the effects of target scaling for DistNet and TabPFN in Sections C.4 and C.5, random forest default adjustments in Section C.6, and SMAC3-based random forest hyperparameter optimization in Section C.7.

## C.1. Predictive Performance across Data Regimes, Metrics, and Benchmarks

The main text reports the predictive performance of all evaluated models across data regimes in Figure 1a, focusing on two representative benchmarks: *clasp_factoring* and *lpg-zeno*. We now present the corresponding results for all seven benchmark

scenarios in Figure 2. This extended comparison evaluates how the quality of the predicted runtime distributions changes with the number of context samples, and it reports all four distributional metrics considered in our study: NLLH, CRPS, Wasserstein distance, and the KS statistic.

The comparison includes TabPFN, TabPFN-Naive, DistNet, and the random forest and Gaussian process baselines. For this experiment, we additionally evaluate a SMAC3-tuned random forest to quantify the potential performance gains of the random forest baseline under hyperparameter optimization. All subsequent experiments revert to the default random forest baseline described in Section A.1.

TabPFN outperforms all baselines across nearly all scenarios and metrics, with the largest gains appearing in small-context regimes. The performance gap is widest under the scale-adjusted NLLH metric, because this metric severely penalizes overconfident errors, which the baseline models are likely more prone to when training data is limited. Similar, although less extreme, trends also appear under the remaining distributional metrics, specifically CRPS, Wasserstein distance, and KS statistic. The consistent ranking across these metrics indicates that the observed gains are not specific to a single likelihood-based objective. Instead, the results suggest that TabPFN systematically allocates probability mass more appropriately over the empirical runtime observations.

The baselines demonstrate the typical behavior for tabular algorithm runtime distribution modeling. The Gaussian process is competitive when the training context is small, but it becomes computationally expensive to fit at larger context sizes due to its cubic time complexity in the number of training samples. In contrast, the random forest baseline is substantially cheaper to train and evaluate, but its predictive performance often plateaus earlier. DistNet benefits from larger training sets, but it generally requires much more data to approach the scores reached by TabPFN in the low-data regime. In several scenarios, TabPFN-Naive is also competitive at small context sizes. Since this variant receives no informative instance features, this result should not be interpreted as evidence that features are unnecessary. Instead, it suggests that a strong prior over one-dimensional target distributions can be useful when only few runtime observations are available.

At the largest context sizes, TabPFN does not improve monotonically for every scenario. In some cases, additional context samples are followed by a degradation in predictive quality. One possible explanation is the limited number of unique problem instances in the benchmark data. Since we have 100 runtime observations for each problem instance, larger contexts are more likely to contain multiple repetitions of the same instance. These repetitions share the same instance-feature vector but can have different runtime values because they correspond to different random seeds. The next subsection studies this effect more directly by removing repeated instance observations from the sampled context (Section C.2).

## C.2. Effect of Removing Repeated Instance Observations

The experiment in Section C.1 evaluates how predictive performance changes with the context size. However, the total context size is determined by two distinct factors: the number of unique problem instances and the number of runtime observations sampled per instance. Repeated runs of a randomized algorithm share the same instance-feature vector but can have different runtimes. Thus, they induce multiple targets for the same input. This is a natural property of runtime distribution prediction, but it differs from the standard supervised tabular regression setting, where many exact duplicate inputs with different targets are less common.

To evaluate whether these repeated instance observations contribute to TabPFN's performance degradation at larger contexts, we repeat the experiment from Section C.1 using deduplicated context sampling. For each context size, we construct a TabPFN variant whose context contains at most one runtime observation per each unique problem instance. We compare this variant against the standard TabPFN configuration, where contexts are sampled from the flattened instance-seed training data without removing repeated instances. Figure 3 reports the resulting comparison across all scenarios and metrics.

The deduplicated results show that much of the performance degradation at large contexts is reduced when repeated instance observations are removed. This does not imply that repetitions are uninformative in general; repeated runs are precisely what define the empirical runtime distribution of a randomized algorithm. However, for TabPFN in this setup, additional unique instances appear to be more useful than multiple runtime samples attached to the same instance-feature vector. A possible interpretation is that dense groups of identical features with different targets make the context less similar to the supervised tabular tasks on which TabPFN was pretrained. In such cases, the model may receive instance-specific runtime noise mainly as target disagreement at identical inputs, rather than as uncertainty associated with a particular problem instance.

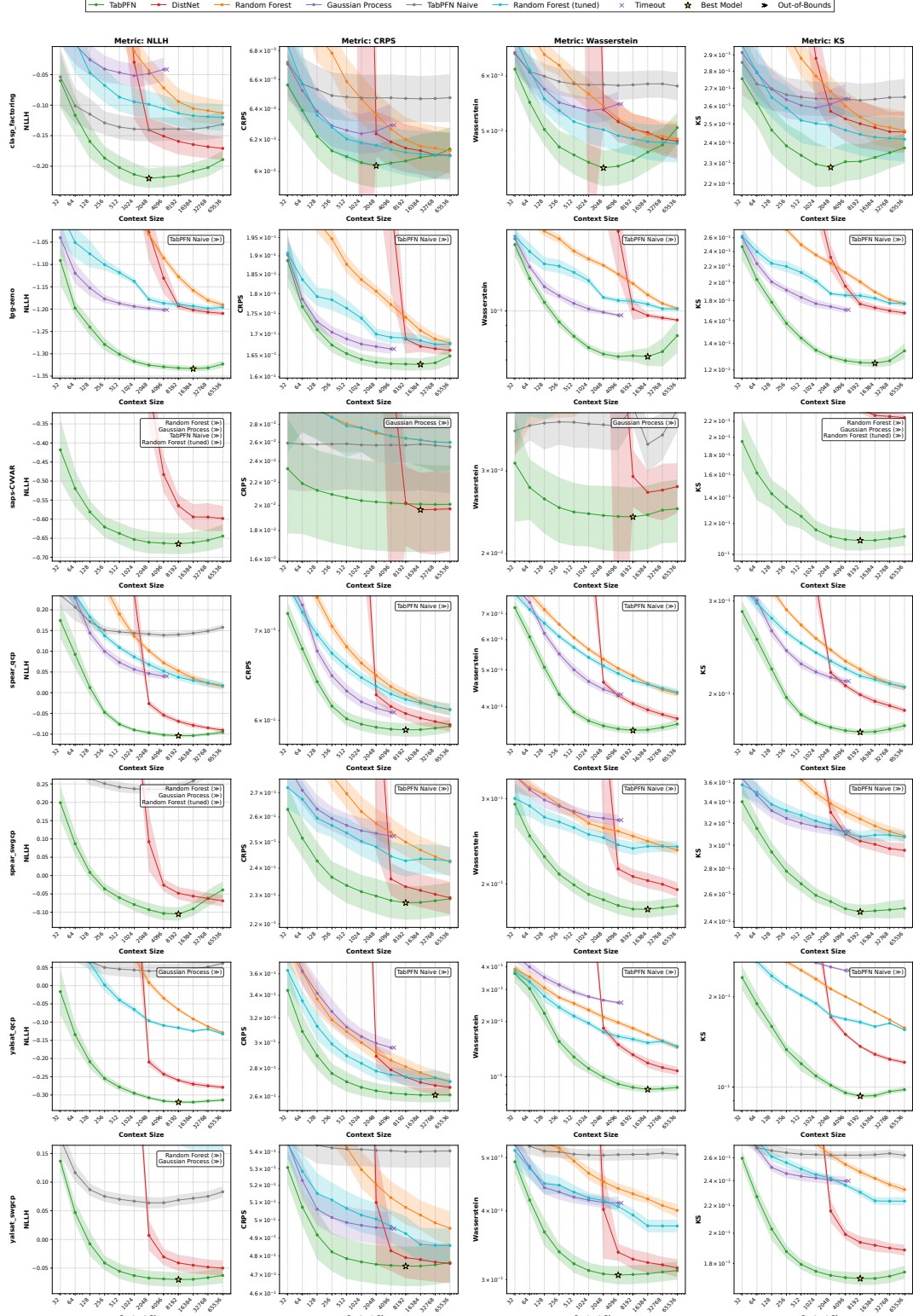

*Figure 2.* **Extended predictive performance across data regimes.** Performance of all evaluated runtime distribution predictors as a function of context size. Rows correspond to benchmark scenarios and columns correspond to NLLH, CRPS, Wasserstein distance, and KS statistic. Solid lines denote mean scores across 10-fold instance-level cross-validation and five random seeds per fold; shaded regions denote ±1 standard deviation. Lower values indicate better performance. Stars mark the best-performing model and context size in each subfigure.

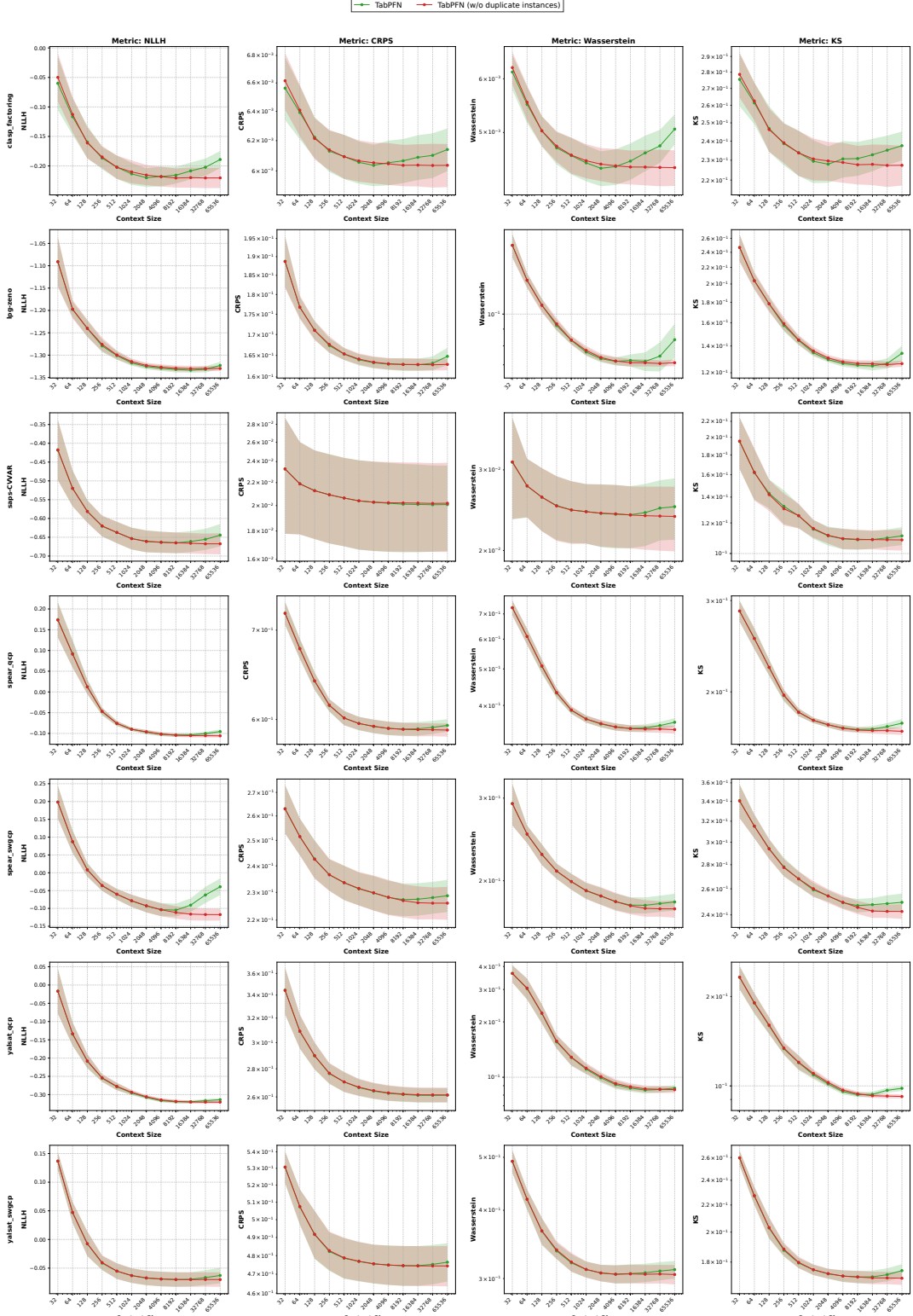

*Figure 3.* **Effect of context deduplication on TabPFN.** Comparison of standard TabPFN and a deduplicated variant across all benchmark scenarios. In the deduplicated variant, the context contains at most one runtime observation per problem instance. Rows correspond to benchmark scenarios and columns correspond to NLLH, CRPS, Wasserstein distance, and KS statistic. Solid lines denote mean scores across 10-fold instance-level cross-validation and five random seeds per fold; shaded regions denote ±1 standard deviation. Lower values indicate better performance.

## C.3. Computational Cost and Memory Scaling

Predictive quality alone does not determine whether a runtime distribution prediction model is practically useful. For algorithm selection, configuration, and portfolio construction, the cost of fitting and querying the model can be part of the overall decision-making budget. We therefore profile the computational cost of all evaluated models as a function of context size. The wall-clock measurements in Figure 4 include fit time, prediction time, and amortized prediction time. Here, amortized prediction time denotes the total wall-clock time spent on fitting and prediction divided by the number of test predictions. For TabPFN, we additionally report peak inference VRAM consumption, as its context-conditioned transformer architecture is the primary source of GPU memory usage.

The timing results show different bottlenecks for different model classes. The Gaussian process is expensive during fitting and repeatedly reaches the one-hour wall-clock limit at larger context sizes. Random forests and DistNet have comparatively low prediction cost once fitted. TabPFN, in contrast, has no task-specific training phase in the usual sense, but its prediction cost increases steeply with the size of the provided context, consistent with the dense-attention computation used by the model. As a result, TabPFN is not the cheapest method per query at larger context sizes, even though it avoids gradient-based optimization on each benchmark.

The amortized prediction time-versus-NLLH trade-off provides a more direct comparison of accuracy and cost, where the most favorable region is the lower-left corner, corresponding to low NLLH and low amortized prediction time. While TabPFN achieves strong NLLH values at moderate context sizes with minimal amortized cost, the memory measurements demonstrate that large contexts demand substantial VRAM. Together with the performance degradation observed at large contexts (Section C.1), these results indicate that simply increasing the context size is an ineffective strategy for algorithm runtime distribution prediction. Instead, more sophisticated context selection, context compression, or instance-level subsampling may be required for the scalable application of PFN-style models to tabular runtime distribution modeling.

## C.4. DistNet Target Scaling Ablation

DistNet is trained by minimizing the negative log-likelihood of a parametric Lognormal predictive distribution, whose scale and shape parameters are predicted by a feedforward neural network. Consequently, the target representation affects both the numerical conditioning of optimization and the scale on which predictive uncertainty is learned. Since runtime observations can span several orders of magnitude, the choice of target scaling is an important modeling consideration.

We compare two target scaling strategies for DistNet in Figure 5: log-transformed runtimes and min–max-scaled runtimes. This ablation uses the same model architecture and evaluation protocol as the main experiments, differing only in the scaling applied to the target variable during training and prediction. We include the min–max-scaled representation to match the original DistNet formulation of Eggensperger et al. (2018).

The min–max-scaled target space yields more stable and generally stronger performance for DistNet across our benchmark suite. We therefore use this variant in the main comparison. This choice should be understood as a baseline-specific implementation decision rather than a general preference for min–max scaling in runtime modeling. For the other probabilistic regression models evaluated, log-space modeling remains more competitive because their Gaussian predictive distributions correspond to Lognormal distributions in the original runtime space.

## C.5. TabPFN Target Scaling Ablation

For TabPFN, the target scaling determines the scale on which the posterior predictive distribution is represented. This is particularly relevant for algorithm runtimes, whose empirical distributions are often heavy-tailed and span multiple orders of magnitude. TabPFN represents its predictive distribution as a binned density over the target variable. If raw or poorly scaled runtimes are provided directly, a large fraction of these bins can be allocated to the largest observed values. As a result, smaller but practically relevant runtime differences may fall into the same or neighboring bins and become harder for the model to distinguish.

We compare two target scaling strategies for TabPFN in Figure 6: log-transformed runtimes and min–max-scaled runtimes. The log transformation yields a target scale on which multiplicative runtime differences become additive, matching the standard modeling practice used by the Gaussian process and random forest baselines. It therefore better preserves relative differences between short and long runtimes than min–max scaling, whose target scale is more strongly influenced by the largest observed values.

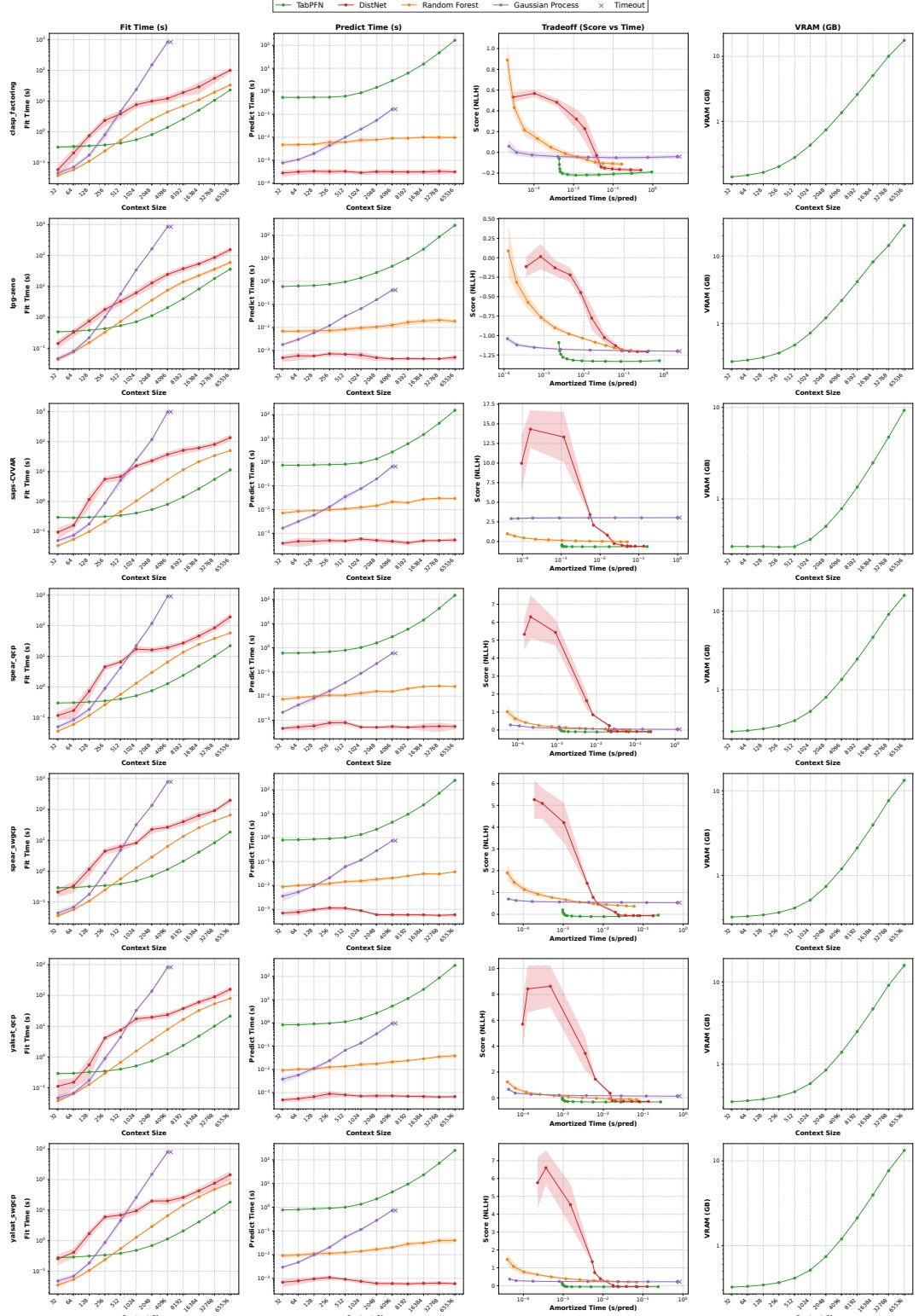

*Figure 4.* **Computational cost and memory scaling.** Computational profiling of all evaluated models across benchmark scenarios (rows). Columns show raw fit time, raw prediction time, amortized prediction time versus NLLH, and peak inference VRAM for TabPFN. Crosses mark runs that reached the one-hour wall-clock limit. Lower time and memory values indicate lower computational overhead; in the amortized time-versus-NLLH subfigures, better trade-offs lie closer to the lower-left corner.

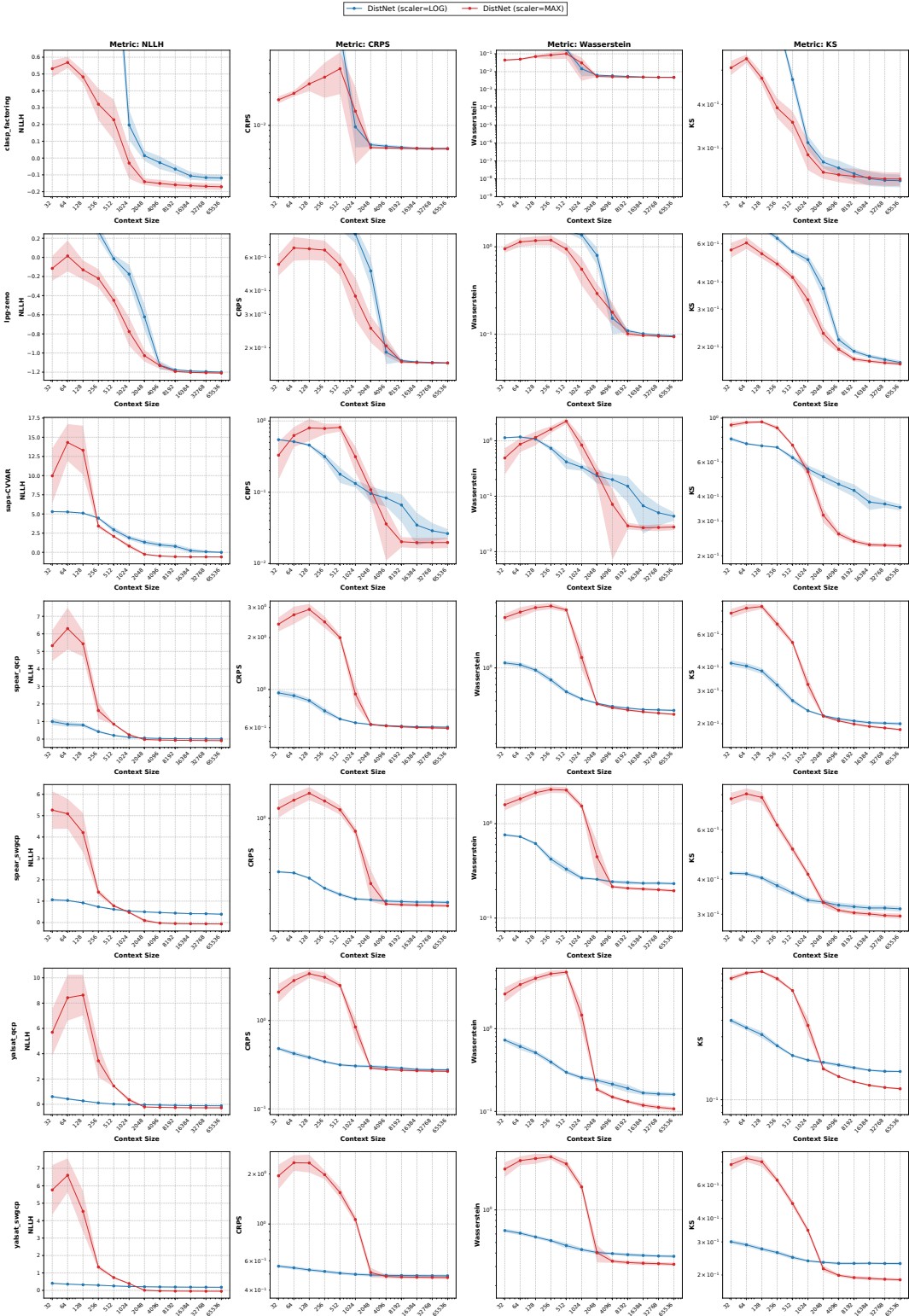

*Figure 5.* **Effect of target scaling on DistNet.** Performance comparison of DistNet trained with log-transformed targets and min–max-scaled targets. Rows correspond to benchmark scenarios and columns correspond to NLLH, CRPS, Wasserstein distance, and KS statistic. Solid lines denote mean scores across 10-fold instance-level cross-validation and five random seeds per fold; shaded regions denote ±1 standard deviation. Lower values indicate better performance.

The results support the log-transformed target scaling used in the main experiments. Across the evaluated scenarios, log-space TabPFN achieves better distributional fit than the min–max-scaled variant. This validates the preprocessing choice made in Section 3 and indicates that the choice of target scaling is an important component of applying TabPFN to runtime distribution prediction.

### C.6. Random Forest Default Configuration Ablation

The random forest baseline is an important point of comparison because random forests are widely used in empirical performance modeling and algorithm configuration (Hutter et al., 2014; Lindauer et al., 2022). We therefore verify that the configuration used in our main experiments is not disadvantaged by outdated defaults. The ablation in Figure 7 compares the original default configuration of Hutter et al. (2014) with the updated configuration used throughout our experiments.

The updated configuration increases the number of trees from 10 to 50 and reduces the minimum predictive variance stored at each leaf from $10^{-2}$ to $10^{-6}$. The larger ensemble reduces variance from finite tree sampling, while the lower variance floor allows the model to represent sharper predictive distributions when the data support them.

The updated configuration yields stronger random forest performance across the evaluated scenarios, so we adopt it throughout the paper. This makes the baseline more competitive without changing the underlying modeling approach: an ensemble of regression trees that predicts a Gaussian distribution in log-runtime space by combining within-leaf variance and between-tree disagreement.

### C.7. Random Forest Hyperparameter Optimization

The manual random forest adjustment in Section C.6 updates a small set of default hyperparameters. To estimate how much additional performance headroom remains for this baseline under systematic tuning, we also optimize the random forest hyperparameters with SMAC3 (Lindauer et al., 2022), using a one-hour wall-clock budget for each tuning run.

The search space includes six hyperparameters: the fraction of features considered at each split (`max_features` $\in [0.4, 1.0]$), the minimum impurity decrease required for a split (`min_impurity_decrease` $\in [10^{-10}, 10^{-3}]$, log scale), the minimum number of samples required to split an internal node (`min_samples_split` $\in \{2, \ldots, 5\}$, log scale), the minimum leaf variance (`var_min` $\in [10^{-6}, 10^{-2}]$, log scale), whether bootstrap sampling is enabled (`bootstrap` $\in \{\text{True}, \text{False}\}$), and the maximum sample fraction when bootstrapping is active (`max_samples` $\in [0.5, 1.0]$). The number of trees is fixed to 50, following the ablation in Section C.6.

SMAC3 optimizes the average validation NLLH on inner folds constructed with three-fold `GroupKFold` at the instance level. This split prevents repeated runs of the same problem instance from appearing in both the inner training and validation folds. Within each inner fold, feature normalization and target log-transformation are fitted on the training portion only. The tuned model is then compared with the updated default configuration in Figure 8.

The tuned random forest improves over the default configuration in most settings, showing that additional tuning effort is useful for this baseline. However, tuning does not change the qualitative role of the random forest in our comparison. It remains a strong classical baseline with relatively low prediction cost, but systematic tuning is not sufficient to close the gap observed between the default random forest configuration and TabPFN in the low-data regimes reported in Section C.1.

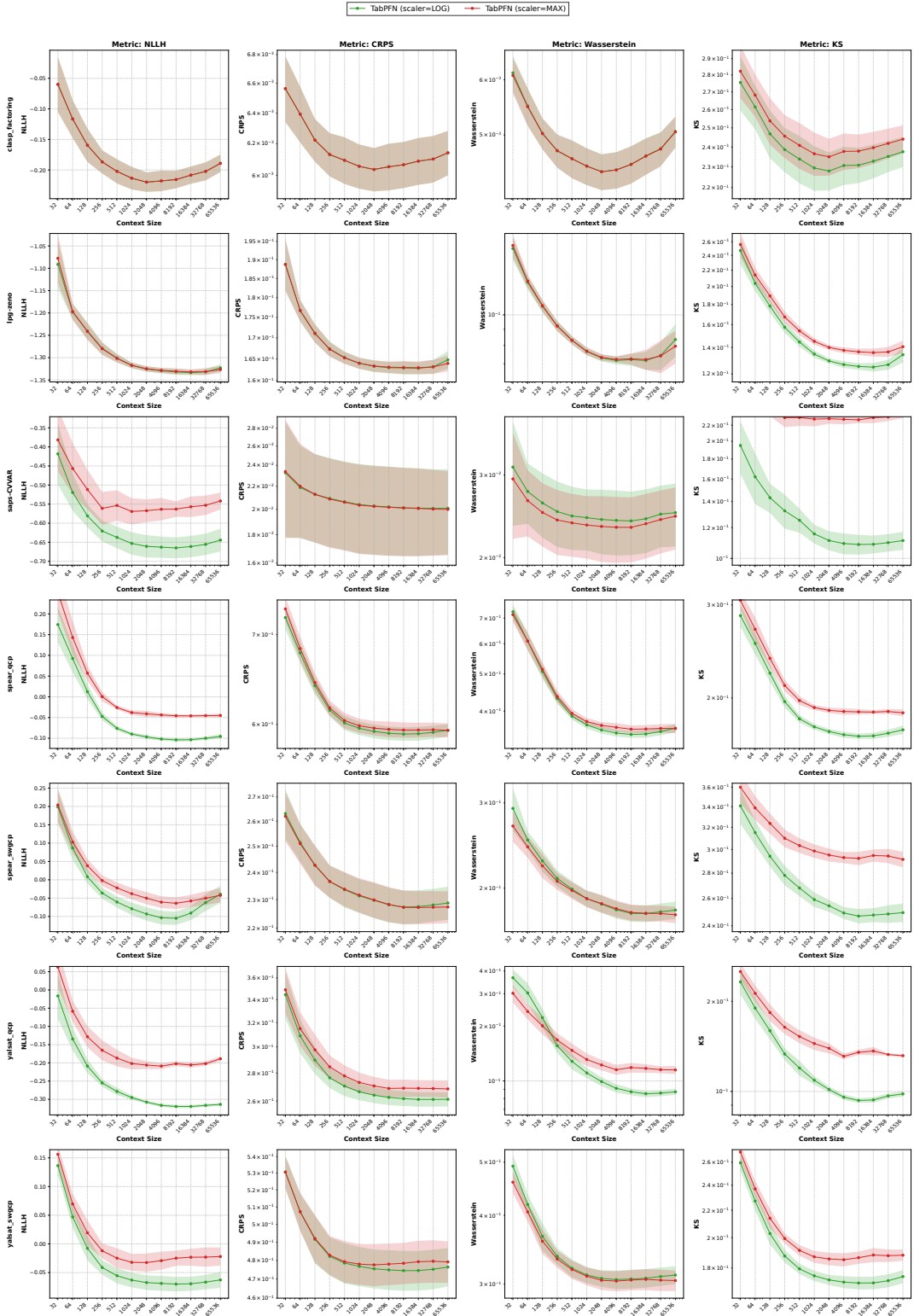

*Figure 6.* **Effect of target scaling on TabPFN.** Performance comparison of TabPFN evaluated with log-transformed runtime targets and min–max-scaled runtime targets. Rows correspond to benchmark scenarios and columns correspond to NLLH, CRPS, Wasserstein distance, and KS statistic. Solid lines denote mean scores across 10-fold instance-level cross-validation and five random seeds per fold; shaded regions denote ±1 standard deviation. Lower values indicate better performance.

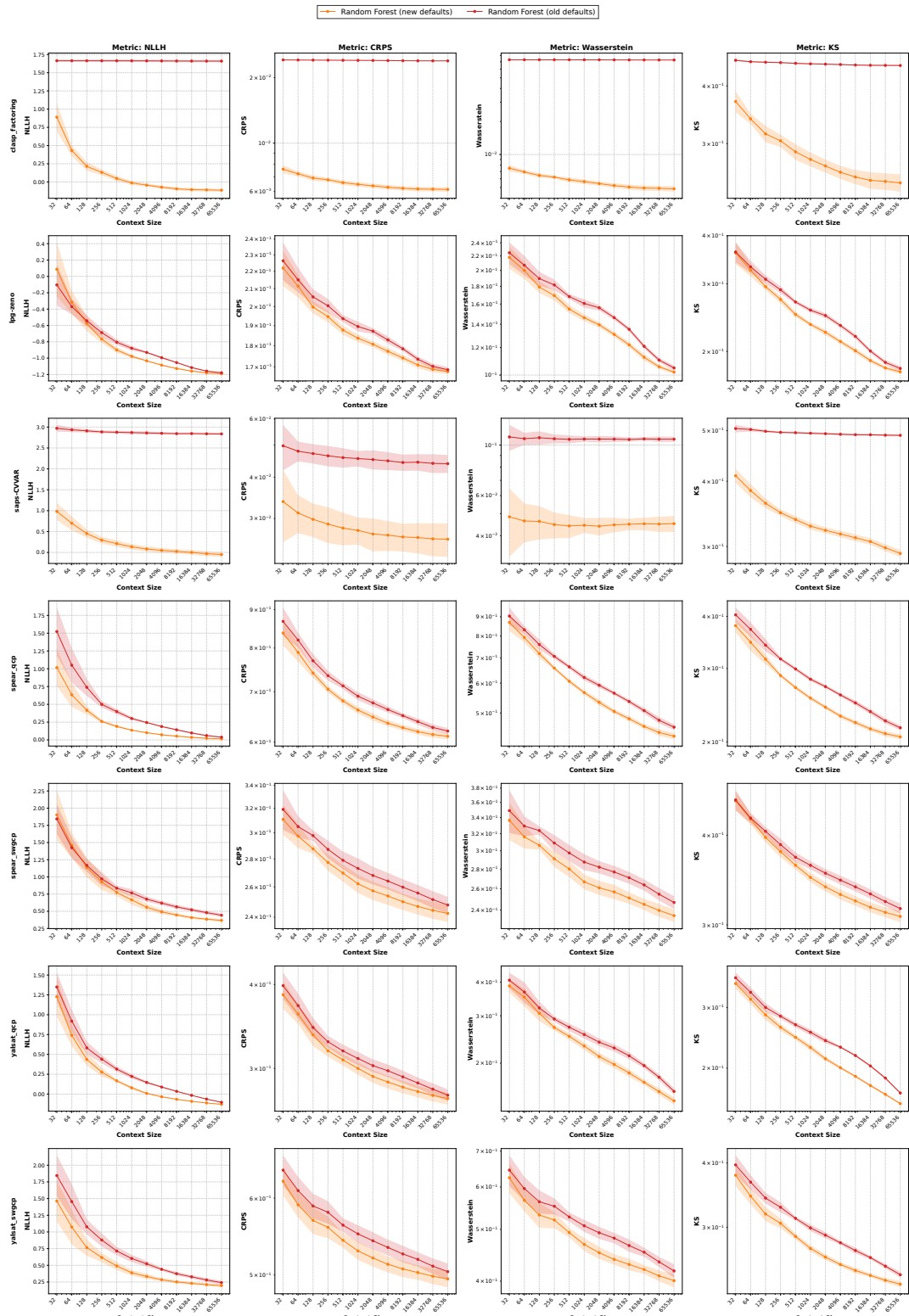

*Figure 7.* **Effect of random forest default adjustments.** Performance comparison between the original random forest default configuration and the updated configuration used in this paper. The updated configuration uses 50 trees and a minimum leaf variance of $10^{-6}$. Rows correspond to benchmark scenarios and columns correspond to NLLH, CRPS, Wasserstein distance, and KS statistic. Solid lines denote mean scores across 10-fold instance-level cross-validation and five random seeds per fold; shaded regions denote $\pm 1$ standard deviation. Lower values indicate better performance.

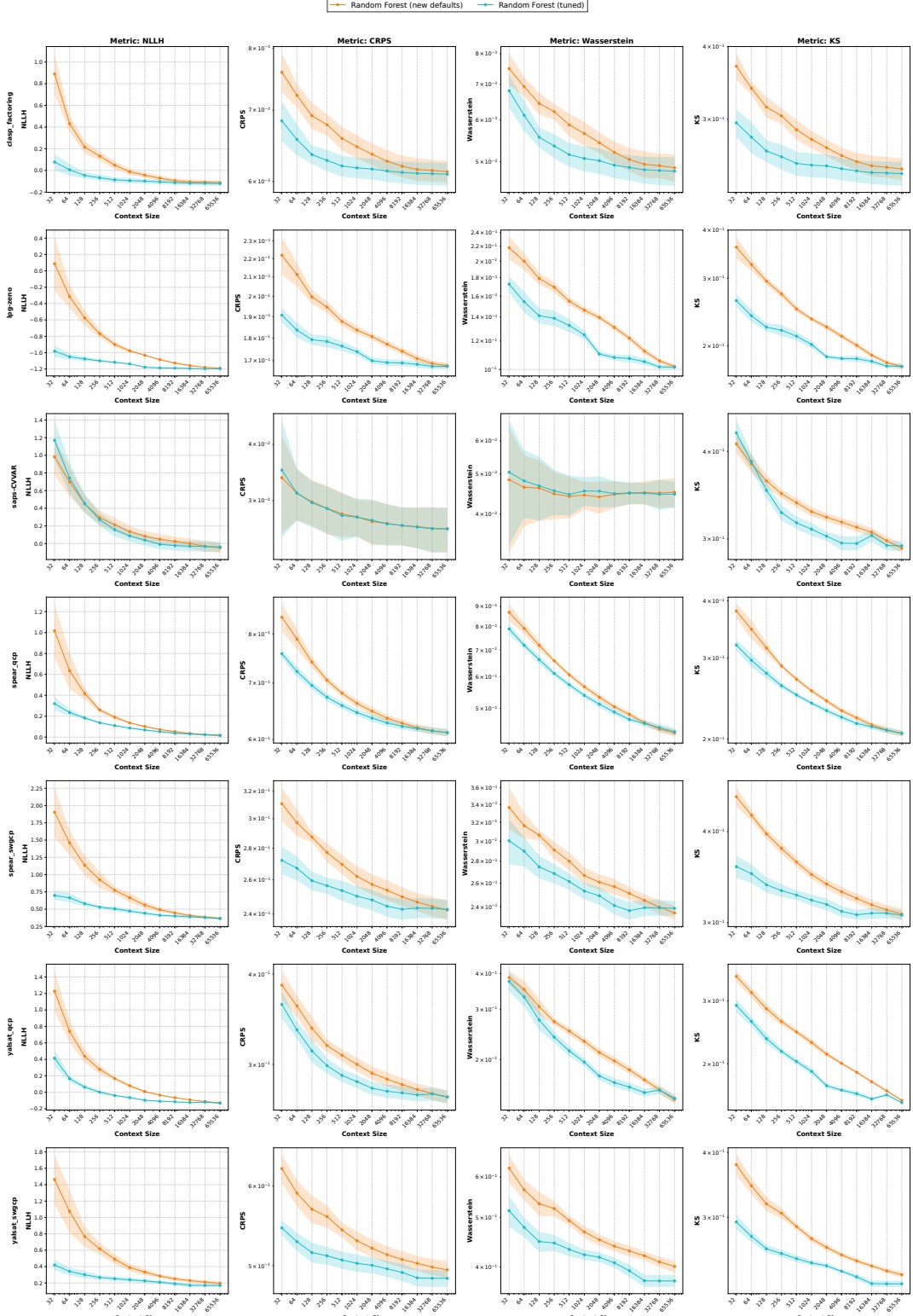

*Figure 8.* **Effect of SMAC3 hyperparameter optimization on random forest performance.** Performance comparison between the updated random forest default configuration and an actively tuned configuration selected with SMAC3. Rows correspond to benchmark scenarios and columns correspond to NLLH, CRPS, Wasserstein distance, and KS statistic. Solid lines denote mean scores across 10-fold instance-level cross-validation and five random seeds per fold; shaded regions denote $\pm 1$ standard deviation. Lower values indicate better performance.

