# OpenReview forum: "Can Tabular Foundation Models Predict Algorithm Runtime Distributions?"
_ICML.cc/2026/Workshop/FMSD — FMSD @ ICML 2026 Poster_

### Official Review · Reviewer_KN3f · 2026-05-15
**Too much focus on the simple evaluations and not the implications.**

**Rating:** 4
**Confidence:** 3

**Review:**

**Summary**
The paper investigates whether TabPFN as a tabular in-context learner can improve and accurately predict algorithmic runtime prediction compared to established baselines such as DistNet or Random Forest. The authors consider both point-wise predictions as well as distirbutions and evaluate them on different metrics. The two main finding of the paper are (1) especially in low-data regimes, TabPFN can outperform the baselines, and (2) TabPFN predictions can become worse with duplications in larger contexts.

**Strengths**
- The problem statement is clearly formulated and the structure of the paper makes sense.
- The figures are well designed, the experiments support the claims.
- The method is described really well for the scope of the paper.

**Areas for Improvement**
- The paper exceeds the 4-page limit of the workshop.
- The framing of the paper suggests that it is merely a "evaluation of  TabPFN on algorithmic runtime distributions". This could be improved by investigating more in a "how can TabPFN improve predicting algorithmic runtime predictions" direction.
- The failure mode of TabPFN for duplications is present in the experiments, but not addressed appropriately in the text.

**Detailed Comments**
-> See Areas of Improvement. Other issues:
- Citations unclear. Oh et al (2023) for instance instead of proceedings name.
- In the appendix (A.3) you describe the metrics used. It would be helpful for the reader to also include the formulas
- ll212: "Future work should address these scalability issues through context compression and alternatives to dense attention" -> What scalability do you mean? The inference time? A solution to the "scalability to larger contexts with duplication" was already proposed in the paper.

**Justification of Score:**
The paper addresses the use of TabPFN for algorithmic runtime distributions, which can be a promising direction. The experiments support the claims. However, the contribution is mostly centered around evaluating how TabPFN performs on 7 different datasets, which is not convincing enough. Also, the paper exceeds the page limit of the workshop. The contribution could have been strengthened by either evaluating more the effects of the more accurate runtime predictions or digging deeper on the failure mode (which is already a known issue in TabPFN).

---

### Official Review · Reviewer_hwCx · 2026-05-20
**Runtime distribution prediction as a testbed for tabular foundation models**

**Rating:** 7
**Confidence:** 4

**Review:**

## Summary

This paper asks whether tabular foundation models can predict algorithm runtime distributions (RTDs), framing runtime prediction as a natural testbed for TabPFN: it is structured, data-limited, and requires calibrated uncertainty estimates. The authors evaluate TabPFN out-of-the-box on seven established SAT and AI planning benchmarks against classical baselines (Random Forest, GP) and DistNet, a neural model designed specifically for this task. TabPFN consistently outperforms all baselines across four distributional metrics (NLLH, CRPS, Wasserstein, KS), with the largest gains in low-data regimes. The paper also uncovers a notable failure mode: performance degrades at larger context sizes due to repeated observations with identical features but varying runtimes, and shows that deduplicating the context largely resolves this.

## Strengths

- Clear and well-motivated problem framing. Runtime prediction is a genuinely useful and underexplored testbed for tabular foundation models, and the paper makes a convincing case for it
- Strong and consistent empirical results. TabPFN outperforms all baselines across all seven benchmarks and all four metrics, with especially compelling gains in low-data regimes that align well with TabPFN's design rationale
- The context degradation failure mode is interesting and well-investigated. The deduplication ablation cleanly supports the repeated-observation hypothesis and is a useful finding for the community

## Areas for Improvement

- Only TabPFN is evaluated among tabular foundation models. The paper acknowledges this but given that TabICL, TabDPT, and others are cited, at least a brief comparison or explanation of why they were excluded would strengthen the positioning
- The context degradation failure mode, while interesting, is somewhat specific to this dataset's structure (100 seeds per instance creating repeated feature vectors). It is worth discussing how representative this regime is of typical TabPFN deployments, and whether this is a known limitation or a new finding

## Detailed Comments

- The framing of runtime prediction as a benchmark for context scaling in tabular foundation models is the paper's strongest conceptual contribution and could be foregrounded more in the introduction
- It would be useful to include a brief summary table of results across all seven benchmarks in the main paper, rather than deferring everything to the appendix. Even a win/loss count across methods and context sizes would help readers grasp the consistency of the findings
- The hypothesis that repeated observations are out of TabPFN's pretraining distribution is plausible but not verified. A brief discussion of what TabPFN's pretraining data looks like with respect to repeated features would strengthen this claim

## Justification of Score

This is a focused, well-executed paper that makes a clear contribution: establishing TabPFN as a strong baseline for runtime distribution prediction and identifying a concrete failure mode with a practical fix. The evaluation is thorough and honest. The main limitation is the single-model focus, but the paper is upfront about this and positions it as a first study.

---

### Official Review · Reviewer_Drwq · 2026-05-20

**Rating:** 5
**Confidence:** 4

**Review:**

The authors evaluate TabPFN v2.5, a pretrained tabular foundation model, on the problem of predicting full algorithm runtime distributions rather than just point estimates. The authors benchmark TabPFN against three baselines: DistNet, a Gaussian Process, and a Random Forest, on seven SAT and AI planning scenarios. They report that TabPFN outperforms all baselines across four distributional metrics (NLLH, CRPS, Wasserstein, KS), with substantial gains in low-data regimes. They also identify that predictive performance degrades beyond a certain context size, which they attribute to repeated observations (same features, different runtimes) being OOD for TabPFN's prior, and show that deduplicating contexts largely flattens this degradation.

Strengths:
1. The problem is well-motivated, and RTD prediction is a genuinely interesting testbed for tabular foundation models owing to small data, structured features, heavy-tailed targets, and a need for calibrated uncertainty.
2. While there exists related PFN work, those works consider related latency or runtime-like settings, but don't compare to TabPFN directly. Filling this gap has value, even if the methodological contribution is to apply TabPFN as is.
3. The results are convincing at a high level, with TabPFN consistently improving over DistNet, RF, and GP across seven benchmarks and four distributional metrics, with large improvements in low-context regimes.
4. The observation that larger contexts can hurt TabPFN, and the follow-up deduplication experiment, is interesting and very pertinent. It points to a real mismatch between TabPFN’s pretraining distribution and repeated noisy observations, which could affect tabular foundation models in general.
5. Using CRPS, Wasserstein, and KS is a real improvement over much of the PFN literature, which often relies on NLLH alone. The justification for using each metric is also relevant and reasonable.

Areas for Improvement:
1. While acknowledged in the paper, there is limited methodological contribution. The paper does not propose a new model, training objective, prior, inference scheme, or runtime-specific adaptation. It is mostly an empirical evaluation of TabPFN on an existing benchmark suite.
2. Only one tabular foundation model (TabPFN) is evaluated, even though the paper claims to ask whether tabular foundation models can predict RTDs. While TabICL, TabDPT, Mitra, and CARTE are all cited in related work, we need experimental results with atleast a couple of more tabular foundation model to support the general claim posed in the title.
3. The deduplication ablation supports the hypothesis, but it also changes the statistical problem. RTD prediction naturally relies on multiple runs per instance. I believe removing repeats avoids the difficulty rather than solving it. A stronger study would test aggregation schemes, noise-aware context construction, or passing seed/randomness information as a feature.
4. Seven benchmarks from one benchmark family are useful but not enough to establish broad conclusions about algorithm runtime prediction. Additional domains, algorithms, cutoff regimes, and feature sets would make the claims much stronger.
5. While the authors show confidence bands, there are no aggregate rankings, Wilcoxon tests, or CD diagrams that can justify the "consistent across all benchmarks" claim well.
6. The authors emphasize that RTDs are heavy-tailed but don't characterize the tail behavior of these benchmarks empirically, nor evaluates tail-specific metrics (like quantile loss at high quantiles).

Overall, this is a competent, honest empirical paper that establishes TabPFN as a strong baseline for RTD prediction and shows a genuinely interesting failure mode (context-size degradation under duplicated observations). However, the methodological contribution is minimal, only one foundation model is tested, which inhibits generalizable claims, and the key empirical claims lack rigorous statistical analysis.